# Numerical Studies on Failure Mechanisms of All-Composite Sandwich Structure with Honeycomb Core under Compression and Impact Loading Conditions

**DOI:** 10.3390/polym14194047

**Published:** 2022-09-27

**Authors:** Xuecheng Han, Hongneng Cai, Jie Sun, Zhiyuan Wei, Yaping Huang, Ang Wang

**Affiliations:** State Key Laboratory for Mechanical Behavior of Materials, Xi’an Jiaotong University, Xi’an 710049, China

**Keywords:** all-composite, honeycomb, compression, impact, failure

## Abstract

The all-composite sandwich structure with the honeycomb core is a lightweight and high-strength structure with broad application scenarios. The face sheet and honeycomb core of the proposed all-composite sandwich structure in this work are composed of carbon-fiber-reinforced polymer (CFRP) composites. The mechanical response and damage mechanism of the all-composite sandwich structure under out-of-plane quasi-static compression and out-of-plane impact are studied by numerical methods. The refined finite element models of the sandwich structures are built on the ABAQUS/Explicit platform. The micromechanics of failure (MMF) theory based on physical component failure is used to describe the intralaminar damage mechanism of the face sheet and honeycomb core, and the mixed-mode exponential cohesive zone model (ECZM) is utilized to simulate the initiation and evolution of interlayer damage. In addition, the cohesive contact approach is adopted to capture the debonding failure at the face-sheet/core. The numerical results show that the all-composite sandwich structure has the characteristics of large structural stiffness and strong energy absorption ability. The failure mechanism of the all-composite sandwich structure under compression is mainly matrix damage and delamination of the honeycomb core, with buckling and folding in appearance. Under out-of-plane impact, matrix damage and delamination arise on the upper sheet, little damage is observed on the lower sheet, and the delamination damage morphology tends to be circular with increasing impact energy. In addition, the interface failure of the upper-sheet/core is more than that of the lower-sheet/core. In addition, the matrix damage near the impact center of the honeycomb core tends to be consistent with the delamination contour, and a small amount of fiber failure is also observed, which manifests as a collapse morphology of the impact area. The research results enrich the understanding of the mechanical behavior of all-composite sandwich structures with honeycomb cores and provide theoretical support for their potential applications.

## 1. Introduction

The honeycomb sandwich structure can achieve high material utilization efficiency and reduce structural weight under the condition of meeting the design requirements of strength and stiffness [1,2,3,4,5,6]. Therefore, it is widely used in aerospace, rail transit, shipbuilding, construction industry, etc. The honeycomb sandwich structure typically consists of two face sheets and a honeycomb core, as shown in Figure 1. The common face sheet materials are aluminum, stainless steel, glass-fiber-reinforced plastic, and composite materials, etc. The typical honeycomb cores include the paper honeycomb, metal honeycomb, and Nomex honeycomb. With the continuous exploration of structural light weight, the design concept of the all-composite sandwich structure with the honeycomb core has been proposed, that is, the face sheets and honeycomb core are composed of fiber-reinforced-polymer composites, which have superior specific strength and specific stiffness. In the life cycle of honeycomb sandwich structures, out-of-plane compression and out-of-plane impact loads are often encountered [7]. As a result, in order to assure the service safety of the all-composite sandwich structure with the honeycomb core, extensive studies on the mechanical response and failure mechanism under the aforementioned loads are required.

Many scholars have conducted a lot of research on the honeycomb sandwich structure from theoretical, experimental, and numerical aspects. Sun et al. [8] studied the indentation and perforation behavior of aluminum honeycomb sandwich panels, and systematically explored the influence of honeycomb structure parameters on quasi-static indentation characteristics, namely, peak force, failure mode, and energy absorption. Gunes et al. [9] investigated the impact properties of all-aluminum honeycomb sandwich panels. Their research showed that the cell size of the honeycomb core has a significant effect on the impact performance, and the change in the core height has no effect on the energy absorption. All-aluminum honeycomb sandwich panels are usually used in the construction and automotive industries. A carbon-fiber-reinforced-polymer (CFRP) composite sandwich panel with an aluminum honeycomb core has higher strength, and it has the function of shielding electromagnetic waves, so it is used in military applications, such as military boxes and shelters. He et al. [10] analyzed the influences of structural parameters such as panel thickness, cell wall thickness, honeycomb core height, and hexagonal side length on the impact response and failure mode of the aluminum honeycomb sandwich panel with CFRP face sheets by combining experiment and numerical simulation. Wu et al. [11] compared the impact properties of CFRP aluminum honeycomb sandwich panels with bare CFRP panels, and described the impact resistance of CFRP structures through energy absorption and impact peaks. Their results show that honeycomb filling is an effective way to improve the impact resistance of CFRP structures. Similarly, Gao et al. [12] evaluated the energy absorption performance of CFRP/aluminum honeycomb sandwich panels and optimized the sandwich structure by numerical simulation. Wang et al. [13] examined the effect of the thickness and density of the aluminum honeycomb core on the mechanical properties of CFRP composite honeycomb sandwich structures. In their research, the three-point bending test was used to assess bending stiffness and strength, while the panel peeling test was performed to estimate the peeling strength. Cai et al. [14] designed the composite sandwich structure with an aluminum honeycomb core, and probed the dynamic response, energy absorption, and damage characteristics under low-velocity impact and high-cycle impact. According to the findings, the honeycomb sandwich structure has a lower peak force, and higher energy absorption and deformation than the single CFRP has. The Nomex honeycomb core has the advantages of light weight, high stiffness, and high strength, so it is frequently utilized in aerospace applications, such as fairings, aircraft doors, and spoilers. Liu et al. [15] carried out tensile, stable compression, and stepwise compression tests to discover the mechanical response of Nomex honeycomb cores under lateral loads. Kim et al. [16] characterized the effect of fluid environment on the mechanical properties of Nomex honeycomb sandwich structures by performing four-point loading tests and impact events. Chen et al. [17] explored the low-velocity impact damage of composite sandwich structures with the Nomex honeycomb core, and used scanning electron microscopy to evaluate the damage. In addition, a numerical model considering intralaminar damage, and interlayer and bonding delamination was established.

It can be seen that researchers have accomplished detailed exploration of honeycomb sandwich structures composed of various materials, which is of positive significance for structural design in different application scenarios. With the development of production technology, many scholars started to explore the preparation process and mechanical properties of all-composite sandwich structures.

Stocchi et al. [18] fabricated a natural-fiber-reinforced composite honeycomb core by vacuum-assisted resin transfer molding. Sugiyama et al. [19] produced all-composite honeycomb sandwich structures using a continuous carbon fiber 3D printer. Wei et al. [20] proposed a tailor-folding method for creating all-CFRP honeycomb sandwich panels. Zhu et al. [21] designed an all-composite sandwich plate with a channel core, which has excellent compressive strength and energy absorption characteristics. In general, unidirectional carbon fiber has better specific strength and specific stiffness. However, there are few reports on a honeycomb sandwich structure composed entirely of unidirectional CFRP composites.

The static mechanical properties of structures are the primary aspect of mechanical behavior analysis. The out-of-plane compression is one of the main static load forms of sandwich structures. Zaharia et al. [22] discussed the performance of lightweight sandwich structures with various core topologies prepared from biodegradable materials through compression tests. The findings revealed that the shear failure is the main failure mode for the sandwich structure. Aktay et al. [23] employed the micromechanical honeycomb model and semi-adaptive coupling technique to simulate the transverse crush behavior of the honeycomb core, and the honeycomb design was guided by the created numerical technique. Sun et al. [24] described the out-of-plane compression characteristics of a hybrid corrugated core sandwich panel. In addition, the sandwich structures are often subjected to impact during service and maintenance. The impact damage characteristics of sandwich structures composed of different materials vary widely. Demirci [25] depicted the low-velocity-impact process of a composite sandwich panel through impact force, displacement, interaction time, and absorption energy. Similarly, Yang et al. [26] investigated the low-velocity-impact response of the sandwich panel with functionally graded carbon-nanotubes-reinforced composite face sheets and a negative-Poisson-ratio auxetic honeycomb core. Xie et al. [27] assessed structural impact damage using nondestructive testing methods such as X-rays, infrared thermography, and ultrasound. Hayta et al. [28] believed that the impact resistance of the composite sandwich structure mainly depends on the core, and the key to the core breakage under impact is the weak binding point of the cell wall. Riccio et al. [29] devised a computational model that can predict the impact behavior of a honeycomb core made of natural fibers, and they analyzed the damage distribution of the fiber and matrix, as well as interlayer damage, during impact. Zhang et al. [30] conducted low-velocity-impact tests on a composite sandwich structure under various impact energies. They observed the damage morphology by an ultrasonic C-scan and optical microscope, and developed a finite element model to fully and clearly demonstrate the panel component damage, the honeycomb core damage, and the core/panel debonding. Palomba et al. [31] also believed that the design of a honeycomb energy absorber requires a broad understanding of its mechanical response under compression and impact loads.

The quasi-static compression performance is an important parameter to evaluate the structure as a bearing component. In addition, the low-velocity impact can cause almost invisible damage in composite structures, which may have catastrophic consequences [32]. As a result, it is necessary to explore the mechanical response and damage mechanism of all-composite sandwich structures with honeycomb cores under quasi-static out-of-plane compression and out-of-plane impact loads. It is worth mentioning that when the honeycomb core is made of unidirectional CFRP composites, the damage mechanism of the honeycomb core becomes more intricate. At present, there are few investigations on the damage mechanism of all-composite sandwich structures with honeycomb cores.

With the rapid rise of numerical computing platforms in recent years, numerical simulation has become a vital tool in the study of the mechanical behavior of materials. The mechanical performance test of composite sandwich structures has the characteristics of long cycle and high cost. As a supplement to the test method, numerical calculation can make the material design more efficient, and illustrate the specifics of material failure clearly.

A new honeycomb sandwich structure composed entirely of unidirectional CFRP composites is proposed. In order to explore the performance characteristics and failure mechanisms of this kind of sandwich structure, this paper focuses on the numerical calculation of the damage evolution of the all-composite sandwich structure under out-of-plane quasi-static compression and out-of-plane low-energy impact loading. In the ABAQUS/Explicit finite element platform, a fine finite element model is established for the out-of-plane compression and out-of-plane impact scenarios of an all-composite sandwich structure with a honeycomb core. The micromechanics of failure (MMF) theory based on physical components failure is constructed to describe the intralaminar damage. The mixed-mode exponential cohesive zone model (ECZM) is used to simulate the initiation and evolution of interlaminar damage. The cohesive contact approach is applied to predict the face-sheet/core debonding. Based on the above theoretical framework, the variation in out-of-plane compressive load with displacement of the all-composite sandwich structure with the honeycomb core is explored, and the deformation and failure mechanisms of the honeycomb core under compression are clearly revealed. The impact dynamic responses such as contact force, impact displacement, and energy absorption under three impact energy levels are studied. In addition, the failure mechanisms of the face sheet, honeycomb core, and face-sheet/core interface are discussed in detail. The findings contribute to a better understanding of the mechanical behavior of all-composite sandwich structures, as well as theoretical advice for prospective applications.

## 2. Analysis Strategy

The all-composite sandwich structure with the honeycomb core has complex geometric features, and it is necessary to establish a calculation method according to the structural characteristics. The MMF theory is used to analyze the mechanical behavior of the intralaminar components. The mixed-mode ECZM is applied to characterize the interlaminar crack propagation. The cohesive surface approach is utilized to capture the damage morphology of the face-sheet/core interface.

### 2.1. MMF Theory

The face sheet and honeycomb core are stacked by unidirectional prepregs, and the MMF theory can be employed to analyze the intralaminar failure. The MMF theory is a multi-scale analysis method based on the components physical failure, which is different from other composite strength theories based on macroscopic strength parameters. The assessment of the intralaminar mechanical state is transformed from the macro to the micro scale, allowing for a more precise understanding of the failure mode and damage evolution of the components. In this theory, the composites are regarded as a continuous homogeneous medium at the macro scale, and the stress amplification factor (SAF) database is built by creating representative volume elements. Then, the micro-stress of the key material points of the fiber and matrix can be computed efficiently, and the failure mechanism of the component can be determined. Figure 2 depicts the MMF theory of CFRP composites. A total of 17 and 19 key points are chosen in the fiber and matrix to represent the stress state of the component, respectively. The numerical link between the micro-stress on the key points and the macro-stress of the element is defined as:(1)$$\sigma_{k}=M_{k}\bar{\sigma }+A_{k}\Delta T$$
where $\bar{\sigma }$ is the macro-stress of the element; $\sigma_{k}$ is the micro-stress of the key point *k*; *M_k_* and *A_k_* are the mechanical stress amplification factor and thermal stress amplification factor of the key point *k*, respectively; *M_k_* is a 6 × 6 matrix; *A_k_* is a 6 × 1 matrix. Δ*T* is the difference value between the curing temperature of the matrix and the room temperature.

The component strength of MMF theory consists of four strength characteristic parameters: tension strength *T*_f_ and compression strength *C*_f_ of the fiber, and tension strength *T*_m_ and compression strength *C*_m_ of the matrix. The strength characteristic parameters of composites used in this work are given in the literature [33].

The component failure criterion of MMF theory is as follows:

Fiber tension failure criterion:(2)$$\max(\frac{\sigma_{11}^{f,k}}{T_{f}})\geq 1$$

Fiber compression failure criterion:(3)$$\max(\frac{\sigma_{\mathrm{VM}}^{f,k}}{C_{f}})\geq 1$$

Matrix tension failure criterion:(4)$$\max(\frac{I_{1}^{m,k}}{T_{m}})\geq 1$$

Matrix compression failure criterion:(5)$$\max(\frac{\sigma_{\mathrm{VM}}^{m,k}}{C_{m}})\geq 1$$
where $\sigma_{11}^{f,k}$ and $\sigma_{\mathrm{VM}}^{f,k}$ are the 1-direction stress component and Mises stress of the key point *k* on the fiber, respectively, and $I_{1}^{m,k}$ and $\sigma_{\mathrm{VM}}^{m,k}$ are the first stress invariant and Mises stress of the key point *k* on the matrix, respectively.

As CFRP composites are orthotropic, the impact of component failure on the mechanical properties of the structure varies substantially. In this paper, the stiffness degradation scheme of CFRP composites is constructed according to the damage variables of components. The flexibility matrix S of CFRP composites after damage is expressed as:(6)$$S=\left[\begin{array}{cccccc} \frac{1}{E_{1}(1-d_{f})} & -\frac{\nu_{12}}{E_{1}} & -\frac{\nu_{13}}{E_{1}} & 0 & 0 & 0\\ & \frac{1}{E_{2}(1-d_{m})} & -\frac{\nu_{23}}{E_{2}} & 0 & 0 & 0\\ & & \frac{1}{E_{2}(1-d_{m})} & 0 & 0 & 0\\ & & & \frac{1}{G_{12}(1-d_{f})(1-d_{m})} & 0 & 0\\ & symmetric & & & \frac{1}{G_{13}(1-d_{f})(1-d_{m})} & 0\\ & & & & & \frac{1}{G_{23}(1-d_{f})(1-d_{m})} \end{array}\right]$$
where *E_i_* (*i* = 1, 2, 3) is the elastic modulus of unidirectional CFRP composites, *G_ij_* (*i*, *j* = 1, 2, 3) is the shear modulus of unidirectional CFRP composites, *ν_ij_* is the Poisson’s ratio of unidirectional CFRP composites, *d*_f_ is the fiber damage variable, and *d*_m_ is the matrix damage variable.

The stiffness matrix of damaged CFRP composites is $C=S^{-1}$, and the expression of ***C*** can be seen in the work [34]. 

For CFRP composites, when the fiber is damaged, the mechanical properties of the fiber decline rapidly, and the matrix damage is usually a gradual damage process. Considering the difference in damage characteristics of component properties, the damage variables of components are written as:(7)$$\begin{array}{l} d_{f}=0.99\\ d_{m}=0.99-0.2\exp[1-{\left(\frac{\sigma_{{}_{S}}}{T_{m}}\right)}^{\lambda }] \end{array}$$
where *λ* is set to 3.5, and $\sigma_{S}$ is the Stassi stress of the matrix [35]. It is worth mentioning that the highest value of *d*_m_ is set to 0.99 in order to prevent element distortion.

In the user-defined material subroutine VUMAT, the degradation of the stiffness matrix is described by updating the damage variables of the components, and then the stress update of the intralaminar components is realized.

### 2.2. Mixed-Mode ECZM

The delamination damage is one of the main failure mechanisms of CFRP composites, resulting in a considerable reduction in bending stiffness and strength. The interlaminar mechanical behavior of the all-composite sandwich structure with the honeycomb core under transverse load is a complicated mechanical process, so the interlaminar cracks are of mixed-mode form. The mixed-mode ECZM is utilized in this research to capture the onset and progression of interlayer cracks. The concept of the mixed-mode ECZM, which considers the impact of tensile and shear stresses on the damage state, is depicted in Figure 3.

The traction–displacement relationship expression of the mixed-mode ECZM is defined as:(8)$$\sigma (\delta )=\frac{G_{C}}{\delta_{m}^{0}}\frac{\delta }{\delta_{m}^{0}}\exp(-\frac{\delta }{\delta_{m}^{0}})$$
where $G_{C}=\sigma_{0}\delta_{m}^{0}e$, $\sigma_{0}$ is the interface strength, δ is the separation displacement, $\delta_{m}^{0}$ is the damage onset displacement, and *e* is a natural constant.

The B–K criterion is employed to describe the energy release rate of epoxy matrix composites in the mixed mode [36]. The expression is written as:(9)$$G_{C}=G_{\mathrm{IC}}+(G_{\mathrm{IIC}}-G_{\mathrm{IC}}){\left(\frac{G_{\mathrm{shear}}}{G_{I}+G_{\mathrm{shear}}}\right)}^{\eta }$$
where $G_{\mathrm{shear}}=G_{\mathrm{II}}+G_{\mathrm{III}}$, and *η* is set to 1.6.

The mixed-mode ECZM is split into two stages: damage onset and evolution. The interlayer damage variable *d*, with a value range of 0~1, is employed to represent the material nonlinear process from damage initiation to complete delamination failure. The damage variables of different crack modes are assumed to be equal in this work. The interlayer damage variable *d* is given by:(10)$$d=1-\exp(1-\frac{\delta }{\delta_{m}^{0}})$$

### 2.3. Cohesive Contact Approach

The out-of-plane dynamic load of the honeycomb sandwich structure will cause debonding at the face-sheet/core interface. The cohesive contact technique is used in this study to model interface debonding failure.

## 3. Finite Element Model

### 3.1. Compression Model

As shown in Figure 4, the numerical model of the all-composite sandwich structure with the honeycomb core subjected to out-of-plane quasi-static compression is constructed on the ABAQUS/Explicit finite element platform. In the numerical model, the all-composite sandwich structure consists of an upper sheet, honeycomb core, and lower sheet. The sandwich structure is placed between two rigid plates. The single-layer thickness of the CFRP layer of the face sheet and honeycomb core is 0.18 mm, the size of the face sheet is 150 × 110 mm^2^, and the stacking sequence of the upper sheet and the lower sheet is consistent, both of which are [0/45/-45/90]_S_. The side wall of the honeycomb core is composed of two CFRP layers, and the straight wall is made up of four CFRP layers, that is, the thickness of the straight wall of the honeycomb core is twice that of the side wall. The side length of the cell is 10 mm and the height of the honeycomb core is 20 mm. The stacking sequence of the straight wall of the honeycomb core is [45/-45/-45/45], and the reference datum of layer orientation is the projection path of the honeycomb core. As the all-composite sandwich structure with the honeycomb core is bonded by unidirectional CFRP prepreg, it can be divided into the intralaminar and interlaminar area, and the unidirectional CFRP performance and interlayer cohesive properties are configured. It is worth mentioning that the interlayer area is set between CFRP layers with varying stacking sequences. The mechanical and thermal properties of the CFRP composites used in this work are shown in [33], and the ECZM parameters of the interlayer are displayed in [34].

In addition, according to the experimental results in the literature [15,37], there is almost no debonding at the interface of the face-sheet/core during the compression process. Therefore, in order to improve the computational efficiency, ‘Tie’ constraints are applied to the interface of the face-sheet/core. All degrees of freedom of the rigid plate below are constrained, and the rigid plate above is subjected to a displacement load. In order to balance the calculation accuracy and efficiency, the compression displacement is set to 2.5 mm, the loading time is set to 20 ms, and the amplitude curve type of the displacement load is set to ‘Smmoth step’. The element type of the CFRP layer is set to C3D8R with enhanced hourglass control. The interlayer thickness is 0.01 mm, and the element type is set to COH3D8.

### 3.2. Impact Model

As shown in Figure 5, the impact model of the all-composite sandwich structure with the honeycomb core is built on the ABAQUS/Explicit finite element platform. The configuration of the all-composite sandwich structure is consistent with the compression model. The boundary conditions refer to the impact test standard of CFRP composite laminates. The all-composite sandwich structure with the honeycomb core is fixed by a support platform and four fixtures. The support plate is a rigid plate with a 125 mm × 75 mm rectangular hole. The impactor weight is 5.6 kg, the end shape is hemispherical, and the diameter is 16 mm. The topological relationship of the interface between the honeycomb core and the face sheet is quite distinct. In order to avoid the element distortion caused by the cohesive element method, the cohesive contact approach is utilized to simulate the initiation and expansion of the debonding of face-sheet/core interface. In addition, in order to facilitate characterization, the upper and lower interfaces are marked as ‘Interface-a’ and ‘Interface-b’, respectively. The interface parameters of the face-sheet/core refer to the literature [17]. In the impact model, the support platform and fixture are defined as rigid bodies, and all the degrees of freedom are constrained. The contact type is defined as ‘general contact’, the normal contact property is set to ‘hard contact’, and the friction coefficient in the tangential contact property is specified as 0.3. The impact response and typical damage mechanism of the all-composite sandwich structure with the honeycomb core under the impact energies of 5 J, 10 J, and 20 J (the corresponding initial velocities of the impactor are 1.335 m/s, 1.889 m/s, and 2.671 m/s) are investigated.

## 4. Results and Discussion

In this section, the dynamic mechanical response and damage evolution process of the all-composite sandwich structure with the honeycomb core under quasi-static out-of-plane compression and out-of-plane impact loading are studied by constructing the numerical calculation framework and the fine finite element models, and some results are compared with the literature conclusions. 

### 4.1. Compression Failure Mechanism

As shown in Figure 6, the failure process of the all-composite sandwich structure with the honeycomb core under quasi-static out-of-plane compression load is presented. Figure 6a presents the compressive-load–displacement curve. Obviously, all-composite sandwich structures have great structural stiffness. It can also be seen that as the loading displacement grows, the compressive load practically linearly increases, and then the payload declines rapidly and loses bearing capacity. Therefore, the quasi-static out-of-plane compression process of the all-composite sandwich structure with the honeycomb core can be separated into three stages, namely the elastic stage, softening stage, and crushing stage. This is similar to the out-of-plane compressive response of other types of honeycomb sandwich structures [38,39]. In order to clearly show the failure process of the all-composite sandwich structure, four key points are selected in the curve to mark the state of the honeycomb sandwich structure, namely A, B, C, and D, and the corresponding compression displacements are 0.11 mm, 0.23 mm, 0.55 mm, and 1.53 mm. During the compression process, the upper and lower face sheets of the all-composite sandwich structure are not damaged, so the associated damage details are not displayed. As shown in Figure 6b–e, when the compression displacement is 0.11 mm, the honeycomb core is in an elastic state and no component damage occurs. When the compression displacement is 0.23 mm, the compression load reaches the maximum value, the side wall of the honeycomb core appears slightly buckled, the matrix begins to be damaged, and most of the damage is distributed on the side wall of the honeycomb core. In addition, the delamination damage is almost nonexistent. When the compression displacement is 0.55 mm, the honeycomb core is in the crushing stage, the honeycomb sandwich structure is unstable, the straight and side walls of the honeycomb core have great buckling failure, a lot of matrix damage occurs, and a large amount of delamination damage is observed on the straight wall, while the delamination damage on the side wall is less. When the compression displacement is 1.53 mm, the honeycomb core shows more significant buckling and folding, the matrix damage is further expanded, and substantial delamination failure emerges on the straight and side walls. In addition, the fiber failure occurs less during compression, so it is not shown. It is easy to see that as the honeycomb core proposed in this work is composed of unidirectional CFRP composites, the failure mechanism of the honeycomb core of the all-composite sandwich structure under compression is more complicated than those of other materials. In summary, under the out-of-plane quasi-static compression, the honeycomb core first shows matrix damage and delamination in the straight wall. As the loading continues, the side wall also experiences delamination. The failure of the honeycomb core is manifested as buckling and folding in appearance. It is easy to find from the numerical results that the damage process of the honeycomb core under out-of-plane compression is gradual and partitioned. These results have important guiding significance for the design of the all-composite sandwich structure with the honeycomb core.

### 4.2. Impact Response and Failure Mechanism

Figure 7 shows the contact-force–time curves of the all-composite sandwich structure with the honeycomb core under impact loads of 5 J, 10 J, and 20 J. The curve comprises an impact and rebound stage. The peak value of the contact force progressively grows with increasing impact energy during the impact process, and the higher the impact energy, the faster the rate of contact force rises. The action time of the impactor and the sandwich structure under the impact energies of 5 J, 10 J, and 20 J is around 7 ms, 8 ms, and 9 ms, respectively, which presents a considerable growth trend. The region enclosed by the curve and the coordinate axis represents the momentum loss of the impactor. It has been discovered that the momentum loss of the impactor increases dramatically as the impact energy increases. In addition, the higher the impact energy, the wider the peak interval.

Figure 8 shows the contact-force–displacement curves of the all-composite sandwich structure with the honeycomb core under 5 J, 10 J, and 20 J impact energies. It can be found that after the interaction between the impactor and the sandwich structure, the upper sheet has obvious dents. The indentation depth is about 0.34 mm, 0.57 mm, and 0.71 mm. The magnitude of the closed curve represents the energy loss of the impact process. The higher the impact energy, the greater the energy loss, indicating that the damage degree of the sandwich structure increases significantly. Furthermore, as impact energy increases, so does the maximum deflection of the sandwich structure.

Figure 9 presents the kinetic energy curves of the all-composite sandwich structure with the honeycomb core under 5 J, 10 J, and 20 J impact energies. The kinetic energy of the impactor is mostly lost during the impact process via matrix damage, fiber breakage, delamination, face-sheet/core interface debonding, and other failure modes. It is obvious that the corresponding kinetic energy losses are 2.58 J, 6.52 J, and 13.57 J, and the corresponding energy loss percentages are 51.6%, 65.2%, and 67.8%. That is, with the increase in the impact energy, the proportion of the kinetic energy loss also increases considerably. The reduction in impactor kinetic energy directly reflects the decline in bearing capacity of the sandwich structure.

Figure 10 illustrates the damage variables of the intralaminar matrix of the upper sheet of the all-composite sandwich structure with the honeycomb core at three impact energy levels. The matrix in CFRP composites plays a role in fixing fibers and transmitting loads. The damage distribution of the matrix has an important influence on the continuous bearing of the sandwich structure. It can be identified that as the impact energy grows, so does the damage area, and the damage distribution is strongly connected to the material direction. It can also be seen that there is severe matrix damage near the center of all the different layers. Furthermore, according to the finite element results, there is almost no component damage in the lower sheet, showing that the sandwich structure has a sufficient protective effect on the lower sheet located on the nonimpact side.

Figure 11 depicts the distribution of interlaminar damage variables of the upper sheet of the all-composite sandwich structure with the honeycomb core under various impact energies, illustrating the damage evolution process of the interlaminar region from damage initiation to macroscopic crack formation. It is easy to find that the distribution orientation of delamination damage tends to coincide with the material direction of the adjacent lower layer. The delamination area near the middle surface of the upper sheet is extensive, which is mainly due to the delamination damage induced by the high shear stress near the middle surface. The interlaminar damage morphology of each layer eventually becomes circular as impact energy increases. It can be inferred that with the increase in impact energy, the delamination damage of each layer will tend to be a circular convergent distribution, rather than an unbounded expansion to the sheet edge. The specific damage area may be related to the geometry of the impactor. Moreover, according to the finite element results, there is no interlayer damage on the lower sheet. The delamination is one of the key damage mechanisms, which will drastically degrade the integrity of the structure, thereby diminishing the bending stiffness.

Figure 12 comprehensively presents the distribution of damage variables of the face-sheet/core interface at various impact energies and impact times, demonstrating the progression of interface damage onset to interface cracking. The damage distribution extends along the contour of the honeycomb hexagon, which is similar to the calculation results of Zhang et al. [30]. The damage of the upper interface occurs before that of the lower interface. The propagation velocity of interface damage along the transverse direction is greater than that along the longitudinal direction. The interface damage of the lower-sheet/core is slightly less than that of the upper-sheet/core. Similarly, interface debonding will destroy the integrity of the structure, and it will also have a significant decline in the continued bearing capacity of the structure.

The honeycomb core of the all-composite sandwich structure is bonded by unidirectional CFRP composites, which have the characteristics of heterogeneity and anisotropy. Therefore, the damage mode is different from the failure mode of homogeneous honeycomb cores in other sandwich structures. The evaluation system based on component failure constructed in this work can clearly show the damage mechanism of the honeycomb core. Figure 13 presents the typical component impact damage state of the honeycomb core of the all-composite sandwich structure. It can be discovered that the area contour of matrix damage and delamination is almost the same. In addition, matrix damage and delamination emerge simultaneously at the same spatial position, which is different from the phenomenon under quasi-static out-of-plane compression. The impact center has obvious buckling phenomenon, which is manifested as folding and collapse. The damage depth and radius rise dramatically as the impact energy increases, as do the matrix damage area and delamination area. A small amount of fiber breakage occurs near the apex of the honeycomb hexagon.

## 5. Conclusions

In this paper, the mechanical behavior of the all-composite sandwich structure with the honeycomb core was investigated by numerical approaches. The typical failure mechanisms of this sandwich structure under quasi-static out-of-plane compression and out-of-plane impact loading were studied. A refined finite element model of the all-composite sandwich structure with the honeycomb core was established, and an analysis framework based on the ABAQUS/Explicit platform was constructed to accurately present the damage distribution. The MMF theory was applied to describe the intralaminar damage mechanism. The mixed-mode ECZM was used to simulate the interlaminar damage initiation and propagation. The cohesive contact method was utilized to model the interface debonding of the face-sheet/core. The deformation process of the all-composite sandwich structure under out-of-plane compressive loading was studied, and the damage evolution of the honeycomb core was presented clearly. In addition, the dynamic responses of the all-composite sandwich structure under the impact energy levels of 5 J, 10 J, and 20 J were researched, and the failure mechanisms of the face sheet, honeycomb core, and face-sheet/core interface were discussed in detail. The research results provide an intuitive and visual reference for the potential application of the all-composite sandwich structure. 

From the analysis of the numerical model, the following conclusions are obtained. First, the structural rigidity of the all-composite sandwich structure is excellent. The damage process of the honeycomb core of the all-composite sandwich structure under out-of-plane quasi-static compression is gradual and partitioned. The main failure mechanisms are matrix damage and delamination, which manifest as buckling and folding. Secondly, the momentum loss increases dramatically as the impact energy increases. The energy absorption percentage of the sandwich structure under the impact energies of 5 J, 10 J, and 20 J is 51.6%, 65.2%, and 67.8%, respectively, that is, the energy absorption effect increases significantly with the increase in impact energy. The impact results show that the main failure modes of the upper sheet are matrix damage and delamination, severe matrix damage occurs at the center of each layer of the upper sheet, and there is a large area of delamination damage near the middle surface of the upper sheet. In addition, with the increase in impact energy, the delamination damage of each layer tends to be a circular convergence distribution state, rather than an unlimited expansion to the sheet edge. There is little damage to the lower sheet. The interface failure of the face-sheet/core expands faster along the transverse direction than along the longitudinal direction. The profile of matrix damage and delamination of the honeycomb core tends to be consistent, and the impact center has obvious buckling, which is manifested as folding and collapse in appearance. Finally, the numerical analysis framework of the all-composite sandwich structure with the honeycomb core constructed in this paper can clearly present its mechanical response and damage details, which provides an effective numerical tool for the application of this kind of sandwich structure.

## Figures and Tables

**Figure 1 polymers-14-04047-f001:**
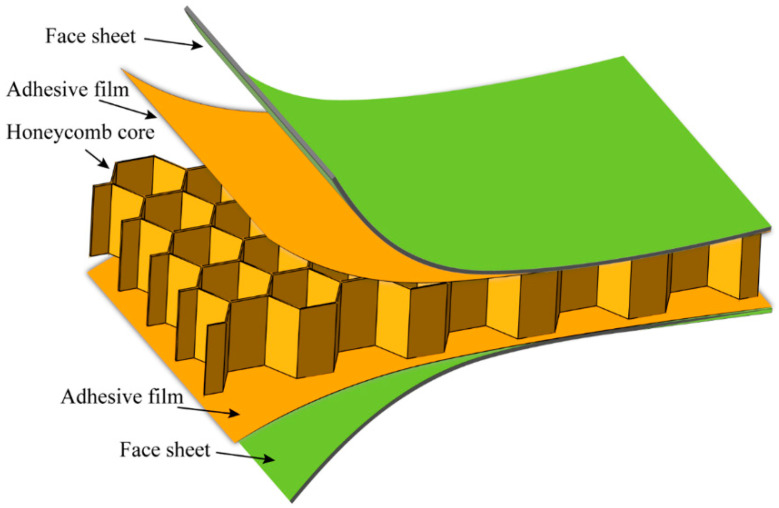
Typical honeycomb sandwich structure.

**Figure 2 polymers-14-04047-f002:**
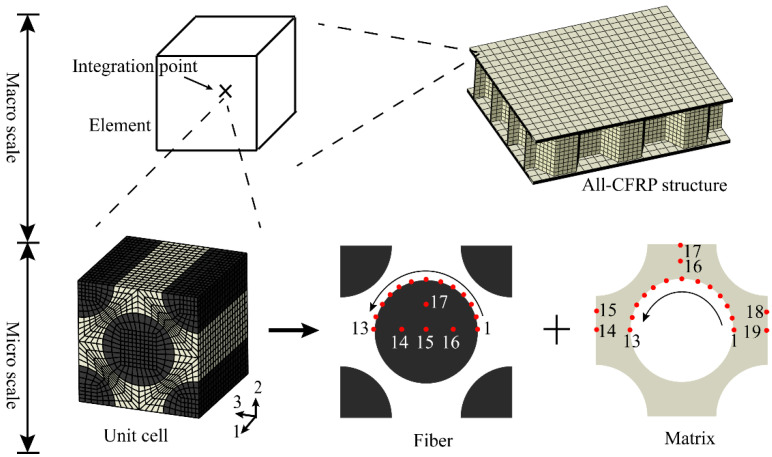
The MMF theory of fiber composites.

**Figure 3 polymers-14-04047-f003:**
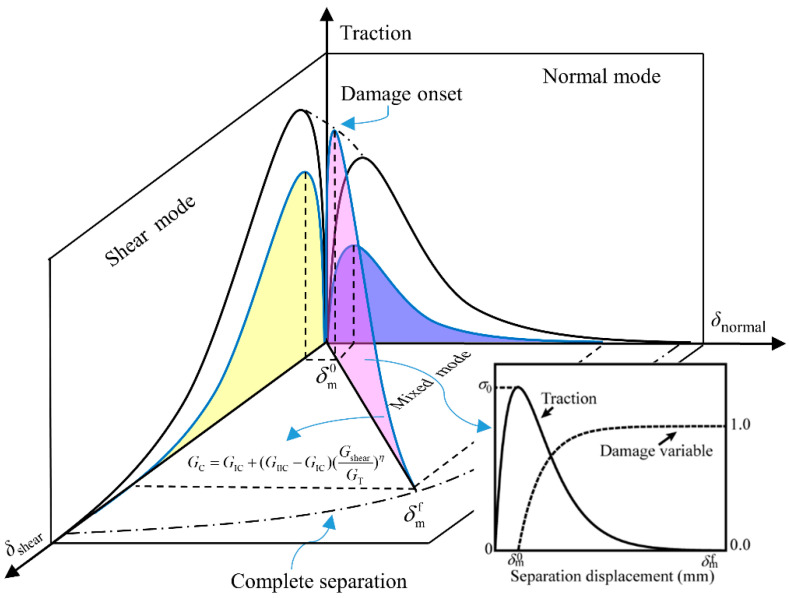
The mixed-mode ECZM.

**Figure 4 polymers-14-04047-f004:**
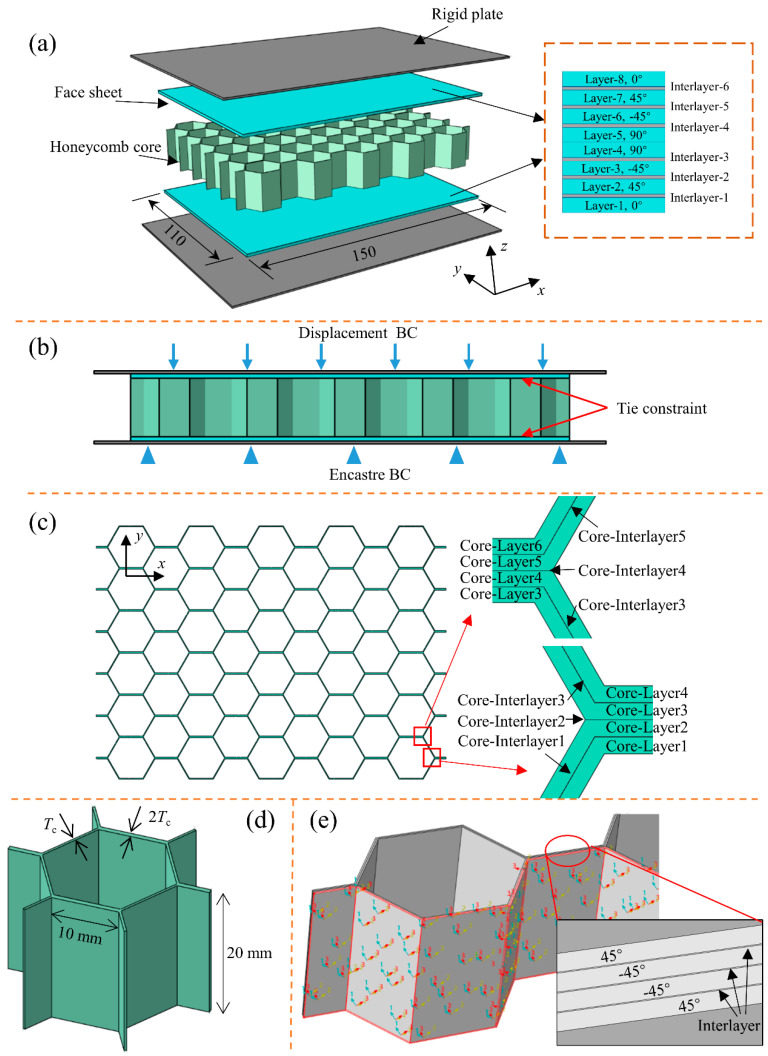
Finite element model of all-composite sandwich structure with honeycomb core under out-of-plane quasi-static compression: (**a**) compression model, (**b**) boundary conditions, (**c**) layer details of honeycomb core, (**d**) cell of honeycomb core, and (**e**) material direction of honeycomb core.

**Figure 5 polymers-14-04047-f005:**
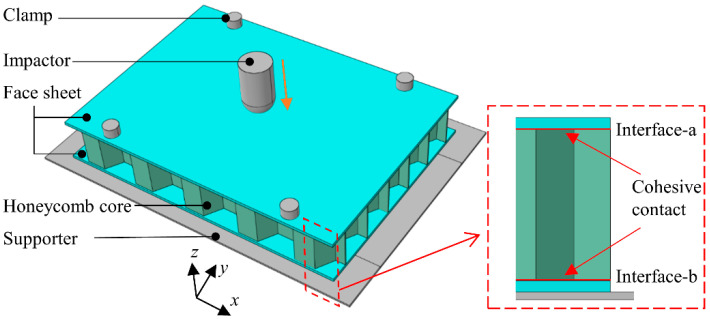
Finite element model of all-composite sandwich structure with honeycomb core subjected to impact.

**Figure 6 polymers-14-04047-f006:**
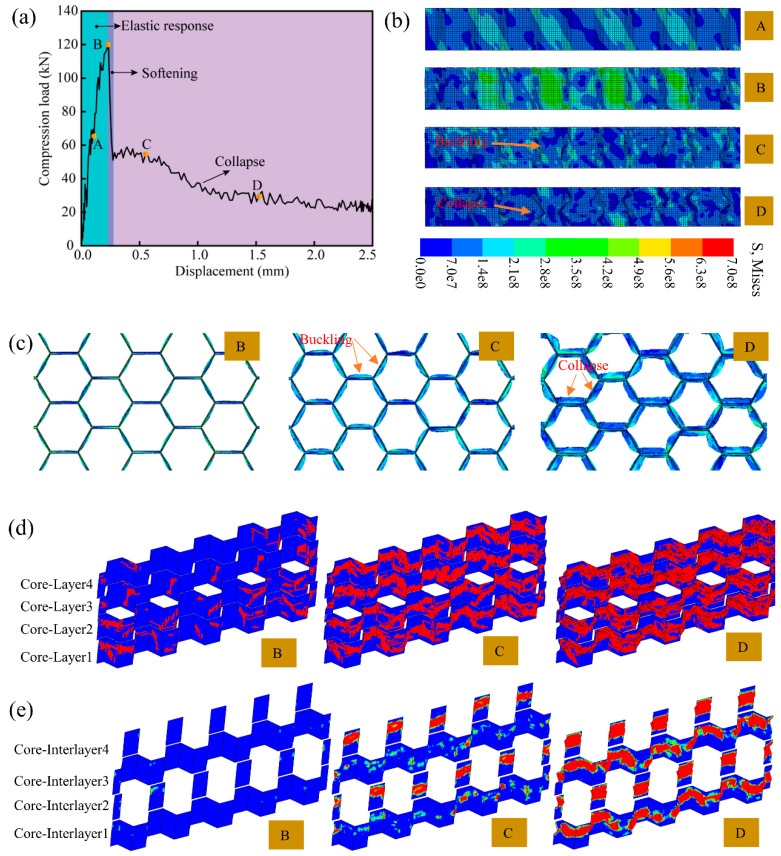
Damage evolution of all-composites under quasi-static out-of-plane compression: (**a**) compression-load–displacement curve, (**b**) the stress contour plot and deformation state of honeycomb core (front view), (**c**) the deformation state of honeycomb core (top view), (**d**) matrix damage distribution of honeycomb core, and (**e**) delamination distribution of the Core-Interlayer1–4.

**Figure 7 polymers-14-04047-f007:**
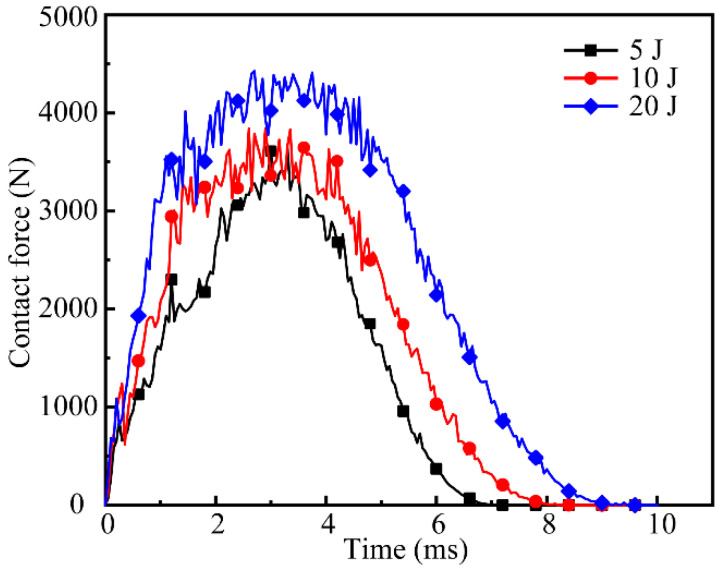
Contact-force–time curves for various impact energies.

**Figure 8 polymers-14-04047-f008:**
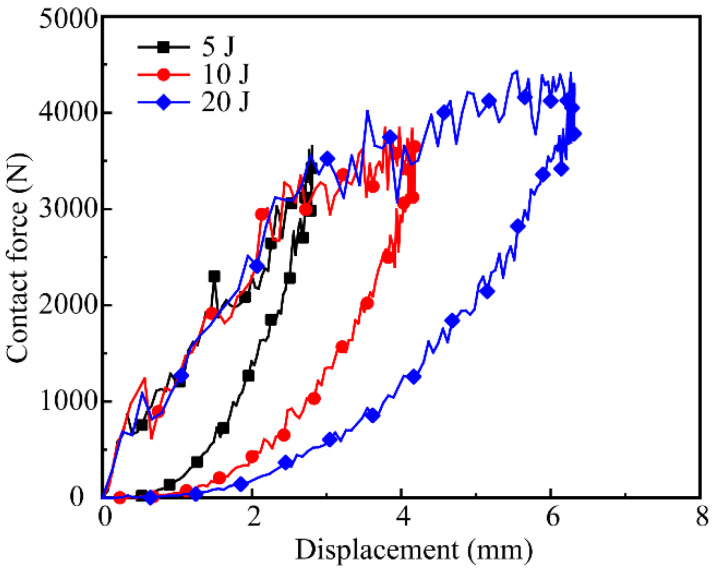
Contact-force–displacement curves for various impact energies.

**Figure 9 polymers-14-04047-f009:**
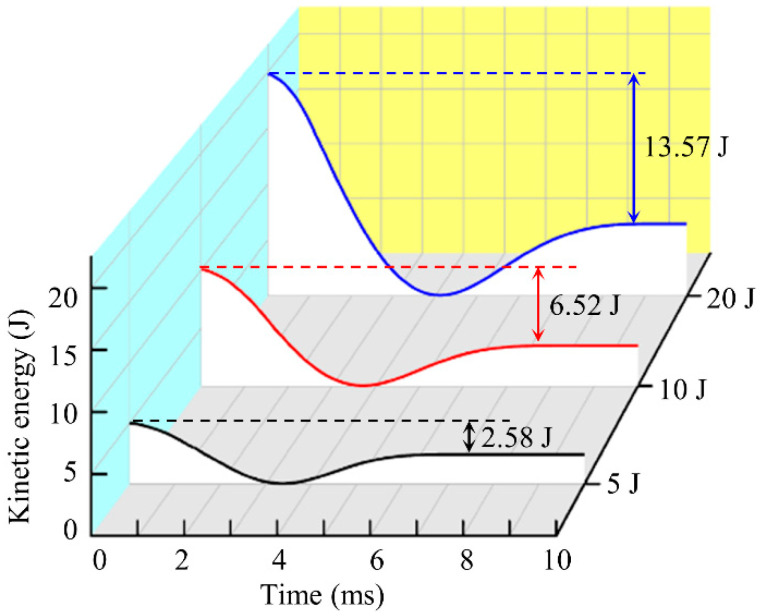
Kinetic energy-time curve of impactor for various impact energies.

**Figure 10 polymers-14-04047-f010:**
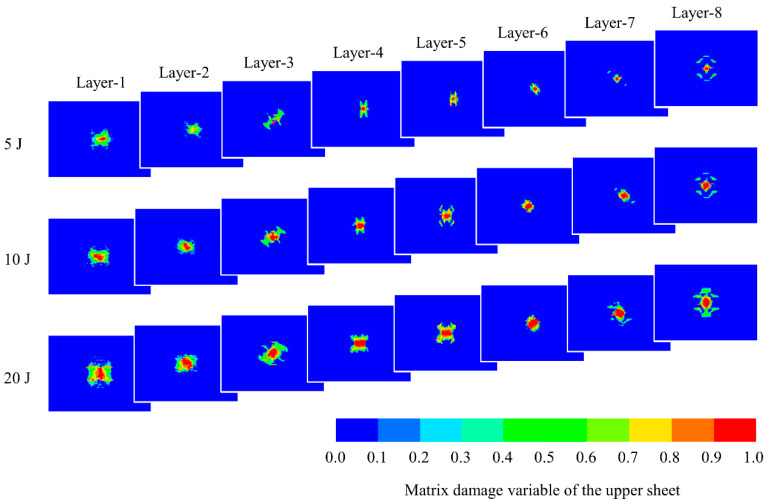
Damage variable of intralaminar matrix in the upper sheet under impact.

**Figure 11 polymers-14-04047-f011:**
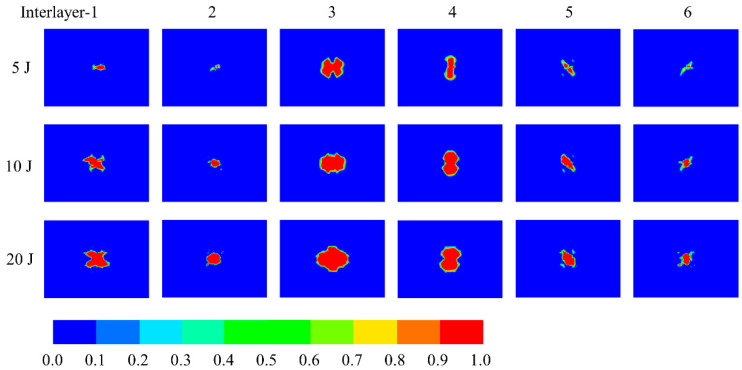
Distribution of interlaminar damage variables of upper sheet under impact.

**Figure 12 polymers-14-04047-f012:**
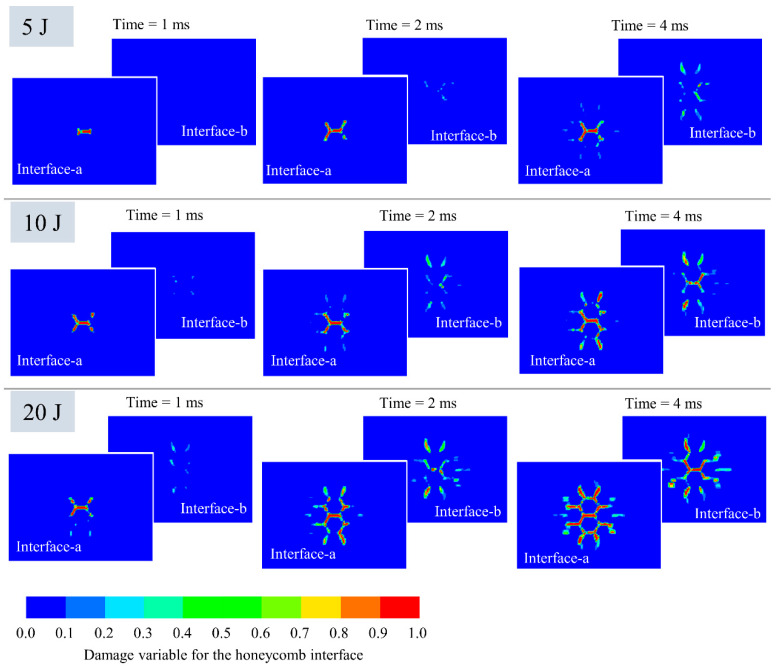
Evolution of interface damage variables of face-sheet/core under impact.

**Figure 13 polymers-14-04047-f013:**
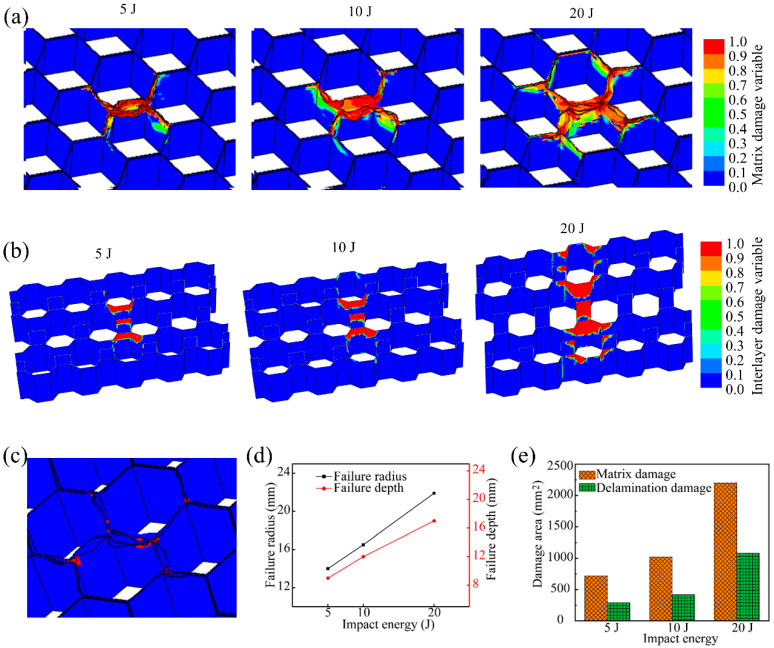
Impact damage of honeycomb core: (**a**) matrix damage variable near the center of honeycomb core, (**b**) delamination damage variable near the impact center of honeycomb core, (**c**) fiber fracture near the impact center of honeycomb core under 20 J of impact energy, (**d**) average failure radius and depth, and (**e**) area of matrix damage and delamination of honeycomb core.

## Data Availability

The data used to support the findings of this study are available from the corresponding author upon request.

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
