# Peer review of "Numerical Studies on Failure Mechanisms of All-Composite Sandwich Structure with Honeycomb Core under Compression and Impact Loading Conditions"

_polymers, 2022, doi:10.3390/polym14194047_

Round 1
Reviewer 1 Report
The authors investigated the failure process of honeycomb sandwich structure during compression and impact loading. The simulation is built in Abaqus, with failure mechanism and mix-mode cohesive zone model are implemented. The model is sound and the conclusion is supported by the data. It is recommended for publication after following clarification questions are answered:
1. Buckling is often deemed as a main structural failure mode for thin wall honeycomb structure. However, it is not significant in the result shown by Fig. 6. What is the main property that governs the buckling failure mode vs. wall interfacial failure?
2. What are the elastic moduli of each layer of the honeycomb wall? I couldn’t find it in the paper yet it is important to analyze the damage initiation and accumulation.
3. What is the design principle of the layer orientation in face sheet? A different layer sequence may result in different reflective index, and therefore different damage pattern. Can the model be used to determine an optimal stack sequence?
Reviewer 2 Report
This paper is a significant study of the Failure mechanism of all-composite sandwich structure with honeycomb core under compression and impact loading conditions. In general the numerical results and graphic are good. However there are some concerns and clarifications need authors attentions are necessary, as: In my opinion, the title may be changed. Also, in the abstract section, please add more results. Please add author recent references in the introduction section and it is necessary to mention clearly the originality and novelty of this study in the end of this section.
